# Learning to Control Visual Abstractions for Structured Exploration in Deep Reinforcement Learning

## Abstract

Exploration in environments with sparse rewards is a key challenge for reinforcement learning. How do we design agents with generic inductive biases so that they can explore in a consistent manner instead of just using local exploration schemes like epsilon-greedy? We propose an unsupervised reinforcement learning agent which learns a discrete pixel grouping model that preserves spatial geometry of the sensors and implicitly of the environment as well. We use this representation to derive geometric intrinsic reward functions, like centroid coordinates and area, and learn policies to control each one of them with off-policy learning. These policies form a basis set of behaviors (options) which allows us explore in a consistent way and use them in a hierarchical reinforcement learning setup to solve for extrinsically defined rewards. We show that our approach can scale to a variety of domains with competitive performance, including navigation in 3D environments and Atari games with sparse rewards.

## 1 Introduction

Exploration in environments with sparse feedback is a key challenge for deep reinforcement learning (DRL) research. In DRL, agents typically explore with local exploration strategies like epsilon greedy or entropy based schemes. We are interested in learning structured exploration algorithms, grounded in spatio-temporal visual abstractions given raw pixels. In human perception and its developmental trajectory, spatio-temporal pixel groupings is one of the first visual abstractions to emerge, which is also used for intrinsically motivated goal-driven behaviors (Spelke & Kinzler, 2007). Inspired by this insight, we develop a new agent architecture and loss functions to autonomously learn visual abstractions and ground temporally extended behaviors in them.

Our approach and key contributions can be broken down into two parts: (1) an information theoretic loss function and a neural network architecture to learn visual groupings (abstractions) given raw pixels and actions, (2) a hierarchical action-value function agent which explores in the space of options grounded in the learned visual abstractions, instead of low level actions.

In the first step, we pass images through an encoder which outputs spatial discrete vector-quantized (VQ`) grids, with 1 of $\mathbf{E}$ discrete components. We train this encoder to maximize the mutual information between VQ layers at different time steps, in order to obtain a temporally consistent representation, that preserves controllability and appearance information. We extract segmentation masks from the VQ layers for the second step, referred to as *visual entities*. We compute affine geometric measurements for each entity, namely centroid and area of the corresponding segment. We use off-policy learning to train action-value function to minimize or maximize these measurements, referred collectively as the *options bank*. Controlling these measurements enable higher levels of behaviors such as approaching an object (maximizing area), avoiding objects (minimize area or minimize/maximize centroid coordinates), moving an object away towards the left (minimize centroid x coordinate), controlling the avatars position on the screen etc.

Finally, given a task reward, we use off-policy learning to train a meta action-value function that takes actions at fixed intervals and selects either one of the policies in the options bank or low-level actions. So effectively, this hierarchical action-value function setup solves a semi markov decision process as in (Sutton et al., 1999; Kulkarni et al., 2016).

We demonstrate that our approach can scale to two different domains – navigation in a 3D environment and challenging Atari games – given raw pixels. Although much work remains in improving the visual and temporal abstraction discovery models, our results indicate that it is possible to learn bottom-up structured exploration schemes with simple spatial inductive biases and loss functions.

## 2 RELATED WORK

Learning visual abstractions has a long history in computer vision with some of the earlier successes relying on clustering either for inference (Shi & Malik, 2000) or learning (Ren & Malik, 2003). More recently some of these intuitions were adapted to neural networks as in (Xia & Kulis, 2017). Instance segmentation algorithms have been developed to output spatio-temporal groupings of pixels from raw videos (Romera-Paredes & Torr, 2016). However, most of the existing deep learning based approaches require supervised data. Structured deep generative models (van den Oord et al., 2017) is another approach to learning disentangled representations from raw videos. Very recently segmentation has been cast as a mutual information maximization problem in (Ji et al., 2018) where the mutual information is computed between the original image segmentation and the output of its transformation (additive color changes in HSV space, horizontal flips and spatial shifts). That approach uses the discrete mutual information estimate which makes it only applicable to enforce pixel-label constraints. For continuous variables recent papers have proposed very promising techniques. Most relevant to our work are (van den Oord et al., 2018) and (Belghazi et al., 2018).

In reinforcement learning research, semi-MDPs and the options framework (Sutton et al., 1999) have been proposed as a model for temporal abstractions of behaviors. Our work is most similar to hierarchical-DQN (Kulkarni et al., 2016). However, this approach required hand-crafted instance segmentation and the agent architecture is not distributed to learn about many intrinsic rewards learners at the same time. Object-Oriented-MDPs (Diuk et al., 2008) uses object oriented representations for structured exploration but requires prebuilt symbolic representations. A recent paper also demonstrates the importance of object based exploration when humans learn to play video games (Dubey et al., 2018). HRA (Van Seijen et al., 2017) is an agent that used prebuilt object representations to obtain state of the art policies on Pacman using object based structured exploration. Another interesting line of work is (Gregor et al., 2016) which formalizes a notion of empowerment which ends up as a mutual information between options in the same MDP. Count-based exploration algorithms have yielded impressive results on hard exploration Atari games (Ostrovski et al., 2017).

The Horde architecture (Sutton et al., 2011) proposed learning many value functions, termed as Generalized Value Functions (GVFs), using off-policy learning. This work was later extended with neural networks by Schaul et al. (2015). Our approach automatically constructs GVFs or a UVFA using the abstract entity based representations. Our work is also related to pixel control (Jaderberg et al., 2016) as an auxiliary task. However, we learn to control and compose abstract discrete representations.

**Notation.** Unless explicitly stated otherwise, we will use calligraphic upper-case letters for sets and MDPs, upper-case letters for random variables, bold capitals for constants and lower-case letters for realizations/measurements.

## 3 MODEL

Consider a Markov Decision Process $\mathcal{M} = (\mathcal{S}, \mathcal{A}, \mathbb{P}, \mathbf{r})$ (MDP) represented by states $s \in \mathcal{S}$, actions $a \in \mathcal{A}$, transition distribution function $\mathbb{P} : \mathcal{S} \times \mathcal{A} \times \mathcal{S} \to [0, 1]$ and an extrinsic reward function defined as $\mathbf{r} : \mathcal{S} \times \mathcal{A} \to \mathbb{R}$. In a discrete MDP, an agent observes a state $s_t$ at time $t$, produces and action $a_t$ then the agent can observe $s_{t+1} \sim \mathbb{P}(S'|S = s_t, A = a_t)$ and a reward $r_{t+1} = \mathbf{r}(s_t, a_t)$. The agent's objective is to maximize the expected sum of rewards over time. In this work we are focusing on visual inputs thus we assume $\mathcal{S} \subset \mathbb{R}^{\mathbf{H} \times \mathbf{W}}$, where $\mathbf{H}, \mathbf{W}$ are the height and width of the image. This is a very important special case in current applications but many of the intuitions and machinery we develop should carry over to different domains.

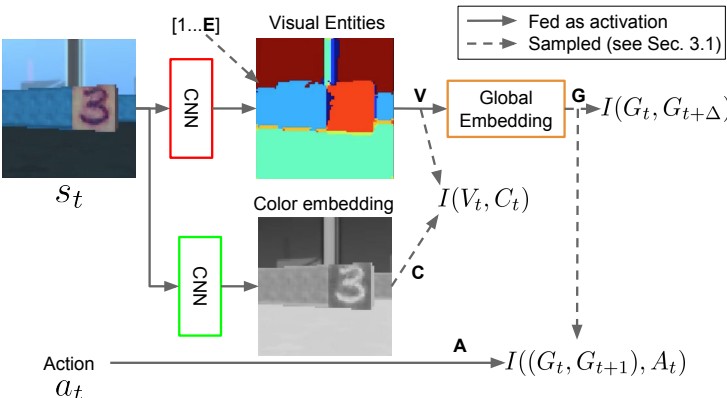

Figure 1: **Unsupervised Visual Abstraction Model:** There are three mutual information losses being computed in this process: $I(G_t, G_{t+\Delta})$, $I((G_t, G_{t+1}), A_t)$ and $I(V_t, C_t)$. $I(G_t, G_{t+\Delta})$ forces the VQ to distinguish frames in the same unroll or large temporal segment with frames outside of this window. $I((G_t, G_{t+1}), A_t)$ encourages the VQ to encode action controllable elements in the frame. $I(V_t, C_t)$ forces the VQ to represent color information.

Our agent also learns an separate abstract representation we call *visual entities*[1] and denote with $v \in \mathcal{V}$, where $\mathcal{V} \subset \{1 \ldots \mathbf{E}\}^{\mathbf{H} \times \mathbf{W}}$ i.e. it assigns an entity id to each location. These are meant to capture useful information about the state $s$ and form the basis for computing the intrinsic rewards $\mathbf{r}^{e,m}$ (see Sec. 3.1). Here $e \in \{1 \ldots \mathbf{E}\}$ denotes a discrete entity id, where $\mathbf{E}$ is the maximum number of possible entities, and $m \in \{1 \ldots \mathbf{M}\}$ denotes a geometric feature of $e$'s derived segmentation mask, from $\mathbf{M}$ possible measurements. In this work the measurements are fixed to what we consider a sufficient set that captures the essential information for natural 3D navigation and Atari game play, and settled on centroid cartesian coordinates and entity mask area, which we can both minimize and maximize, thus in all our experiments $\mathbf{M} = 6$. Temporal changes in these measurements constitute our intrinsic reward functions[2] which induce $\mathbf{E} \times \mathbf{M}$ additional MDPs $\mathcal{O}^{e,m} = (\mathcal{S}, \mathcal{A}, \mathbb{P}, r^{e,m})$. These intrisic rewards will induce behaviors which should hopefully provide structured exploration in the original MDP i.e. picking a random $Q^{e,m}$ is more likely to lead to higher reward than epsilon greedy exploration.

Our agent architecture tries to leverage this additional structure. The top level MDP $\mathcal{M}$ is represented by $Q^{\text{meta}}$ which outputs action $a_t^{\text{meta}}$ at time $t$ (switched every $\mathbf{T}$ time steps), where the action space is discrete 1 of $(\mathbf{E} \times \mathbf{M}) + 1$ possible actions. In our implementation this is modelled by composite actions $\mathbf{E} + 1$ and $\mathbf{M}$ i.e. $a_t^{\text{meta}} = (e_t, m_t)$ with $e_t \in \{1 \ldots \mathbf{E} + 1\}$ and $m_t \in \{1 \ldots \mathbf{M}\}$. We also learn $(\mathbf{E} \times \mathbf{M}) + 1$ separate Q functions: $(\mathbf{E} \times \mathbf{M})$ that each solve one of the $\mathcal{O}^{e,m}$ and one for the original MDP $Q^{\text{task}}$. These Q functions are defined over the environment action set and differ only in the reward function (see Figure 2 for schematic representation).

## 3.1 VISUAL ABSTRACTIONS

Our agent relies on an abstraction model that assigns each pixel in the image to one of $\mathbf{E}$ separate abstractions or entities. To obtain this representation, the image is passed through a convolutional network (CNN) encoder to output a spatial grid of the same resolution as the original image. Then a vector quantized layer $V$ (van den Oord et al., 2017) assigns the planar activations to 1 of $\mathbf{E}$ entities. From the agents perspective this means all the pixels that are being grouped together become indistinguishable thus providing the visual abstraction we desire.

Let us define $f : \mathbb{R}^{\mathbf{H} \times \mathbf{W}} \to \{1 \ldots \mathbf{E}\}^{\mathbf{H} \times \mathbf{W}}$ the function that takes the observation $s_t$ at time $t$ and computes an abstract representation corresponding to it $v_t = f(s_t)$. The key question is how to train $f$. One way is to make it representative of the current state. In most of the current literature, this is

---

[1]We use visual entities and visual abstractions interchangeably. Both terms capture the most important qualities of the representation: 1) emergent nature and 2) discreteness.

[2]If the mask is empty then the centroid is $(0, 0)$ and area is zero.

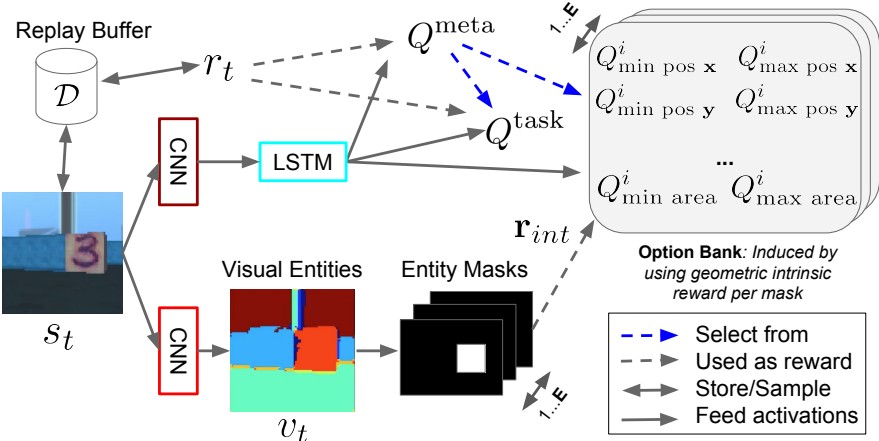

Figure 2: **Agent Architecture:** **(a)** The input $s_t$ is used to compute the *visual entities* $v_t$. Their one-hot encoding is a set of $\mathbf{E}$ $\{0, 1\}$ masks. These are used to compute geometric measurements such as: area and centroid positions (Sec. 3). Temporal differences in these constitute intrinsic rewards $\mathbf{r}_{int}$ for an option bank. **(b)** The input is separately passed through different a CNN and LSTM network, whose output is then fed to: $Q^{\text{task}}$, $Q^{\text{meta}}$ and options bank with $\mathbf{E} \times \mathbf{M}$ Q functions. $Q^{\text{meta}}$ and $Q^{\text{task}}$ are both trained with external task reward but the options bank is trained with the previously computed measurements $\mathbf{r}_{int}$. To act, $Q^{\text{meta}}$ outputs a new action every $\mathbf{T}$ steps. Its actions correspond to selecting and executing either: (1) one of the $Q$ functions in the options bank or (2) the $Q^{\text{task}}$ policy. The selected $Q$ function is then used to produce the actions returned to the environment. All Q functions are trained simultaneously, off-policy, from a shared replay buffer $\mathcal{D}$.

achieved by training a function $g : \{1 \dots \mathbf{E}\}^{\mathbf{H} \times \mathbf{W}} \to \mathbb{R}^{\mathbf{H} \times \mathbf{W}}$ such that $g(f(s_t)) \simeq s_t$. This ensures that the representation preserves *all* the information about the state which is a *sufficient* condition for the purpose. It is not hard to see however that it potentially wastes model capacity on unimportant factors of variation. In this work we aim instead for a *necessary* condition for constraining the representation. To this end we train $f$ to be an injective function i.e. such that a decoding function $g$ can distinguish between states by looking at the representation $\forall x \neq y \implies g(f(x)) \neq g(f(y))$.

Our approach is to formulate classification losses to promote *distinguishability* between different states at different levels of the representation. This was shown, in works like (Barber & Agakov, 2004; Belghazi et al., 2018; van den Oord et al., 2018), to be equivalent to maximizing a lower-bound mutual information between random variables of the representation[3]. By choosing the appropriate random variables and sampling strategies from the joint distribution and marginals we can specify the right representation invariances as follows (see Figure 1 for an overview):

**Preserving global information.** The main term driving the representation learning is a global information term. To estimate it, the VQ layer output at time $t$, $v_t$, is further processed by another CNN encoder to output a frame level embedding vector $G_t$. We train a non-parametric classifier to distinguish pairs of frames from the same trajectory from pairs of frames from different trajectories. For that we form pairs $(g_t, g_{t+\Delta})$, where $\Delta$ is sampled randomly, from pairs $(g_t, g')$, where $g'$ is sampled randomly from a different trajectory. Training this classifier lets us indirectly maximize a lower-bound on $I(G_t, G_{t+\Delta})$. This forces $V$ to preserve enough information that distinguishes this particular trajectory from other ones which tends to remove all irrelevant information like textures and unchanging "background" elements and preserve useful moving elements. Note that as long as predictions are stable across time there is no pressure exerted by this cost to simplify the representation.

---

[3]We write our losses as $I$ because of this fact but this is an approximation.

**Preserving controllable information.** Secondly, we want the controllable information to be preserved in the abstract representation. That is, we want to know what aspects of the input were changed as a result of the action. We achieve this by training to predict which action was taken in a particular transition based on our representation. For that we add another loss, denoted by $I((G_t, G_{t+1}), A_t)$, that maximizes a lower-bound on mutual information between the a pair of consecutive frames and the action that was taken in the transition.

**Preserving local appearance information.** Finally, for hard exploration Atari games, where inputs change very little e.g. Montezuma's Revenge much of the initial experience is from the first room, we add an appearance term[4]. This term is meant to align the abstract representation with appearance changes i.e. promote abstractions have consistent colors. For that we feed the input image into a shallow CNN encoder which outputs an embedding $C$ of the local color and texture structure of the image. We would like for the abstract representation to follow the appearance changes thus we maximize a lower bound on mutual information $I(V_t, C_t)$ in the same way as before. The positive pairs spatially aligned $V$ and $C$ embeddings, and the negative pairs are obtained by sampling appearance embeddings from other spatial locations and pairing them with non-sampled $V$. This promotes $V$ representations that represent well appearance under the representational constraints of the VQ representation.

To learn the agents' abstract representation we minimize a weighted sum of these classification losses

$$\mathcal{L}_{abs}(\theta_{abs}) = \sum_t -\alpha_g \mathbb{E}_{\substack{g_0 \sim \mathbb{P}(G'|g_t) \\ g_i \sim \mathbb{P}(G)}} \log q_g(Z_g = 0|(g_0, \ldots, g_K), g_t) \tag{1}$$

$$-\alpha_c \log q_c(a_t|(g_t, g_{t+1})) - \alpha_a \mathbb{E}_{\substack{c_t^i \sim \mathbb{P}(C_t) \\ v_t \sim \mathbb{P}(V_t)}} \log q_a(Z_c = 0|c_t^i, v_t) \tag{2}$$

where we denote by lower case letter the samples from the corresponding uppercase random variables. $\mathbb{P}(G)$ is the time independent marginal distribution over $G_t$'s. We model $q_g$ as

$$q_g(Z_g = 0|(g_0, \ldots g_K), g_t) = \frac{\exp \phi(g_0, g_t)}{\sum_{i=0}^{K} \exp \phi(g_i, g_t)} \tag{3}$$

with $\phi(., .)$ being the cosine similarity over the embeddings and $K$ is a hyper parameter denoting the number of samples. We can similarly derive the third term. This is possible because the embeddings have the same dimensionality. The second term is a cross entropy based action classifier.

## 3.2 Behavior Abstraction

The agent is represented primarily by three sets of Q functions: $Q^{\text{meta}}$, $Q^{\text{task}}$ and $\{Q^{1,1}, \ldots Q^{e,m}\}$ from the options bank. Each $Q$ function has a corresponding policy we denote by $\pi$, in our experiments these are epsilon greedy policies with respect to the corresponding Q function. We denote $\mathbf{T}$ to be the fixed temporal commitment window for $Q^{\text{meta}}$, which means that it acts every $\mathbf{T}$ steps. Note that $Q^{\text{task}}$ and $Q^{e,m}$ act at each environment time step. We can express all three Q functions as: (1) $Q^{\text{meta}}(s, (e, m)) = \mathbb{E}\left[\sum_{t'=t}^{\infty} \gamma^{t'-t} r_{t'} | s_t = s, a_t^{\text{meta}} = (e, m), \pi^{\text{meta}}\right]$, (2) $Q^{\text{task}}(s, a) = \mathbb{E}\left[\sum_{t'=t}^{\infty} \gamma^{t'-t} r_{t'} | s_t = s, a_t^{\text{task}} = a, \pi^{\text{task}}\right]$ and (3) $Q^{e,m}(s, a) = \mathbb{E}\left[\sum_{t'=t}^{\infty} \gamma^{t'-t} r_{t'}^{e,m} | s_t = s, a_t = a, \pi^{e,m}\right]$.

We represent each Q function with a deep Q network (Mnih et al., 2015), for instance $Q^{\text{task}}(s, a) \approx Q^{\text{task}}(s, a; \theta_{task})$. Each $Q \in \{Q^{\text{task}}, Q^{\text{meta}}, Q^{1,1}, \ldots, Q^{e,m}\}$ can be trained by minimizing corresponding loss functions $- \mathcal{L}_{task}(\theta_{task}), \mathcal{L}_{meta}(\theta_{meta})$ and $\{\mathcal{L}_{bank}^{1,1}(\theta^{1,1}), \ldots, \mathcal{L}_{bank}^{e,m}(\theta^{e,m})\}$. We store experiences $(s_t, (e_t, m_t, a_t), r_t, s_{t+1})$ in $\mathcal{D}$, a shared buffer from which all the different $Q$ functions can sample to perform their updates. The transitions are stored in such a way as to be able to sample trajectories.

The Q-learning (Sutton & Barto, 1998) objective can be written as minimizing the loss:

$$\mathcal{L}_{task}(\theta_{task}) = \mathbb{E}_{(s_\tau, a_\tau, r_\tau, s_{\tau+1})_\tau \sim D}[(R_\tau - Q^{\text{task}}(s_\tau, a_\tau; \theta_{task}))^2] \tag{4}$$

---

[4]We found that for tasks that have richer visual variation this was not necessary.

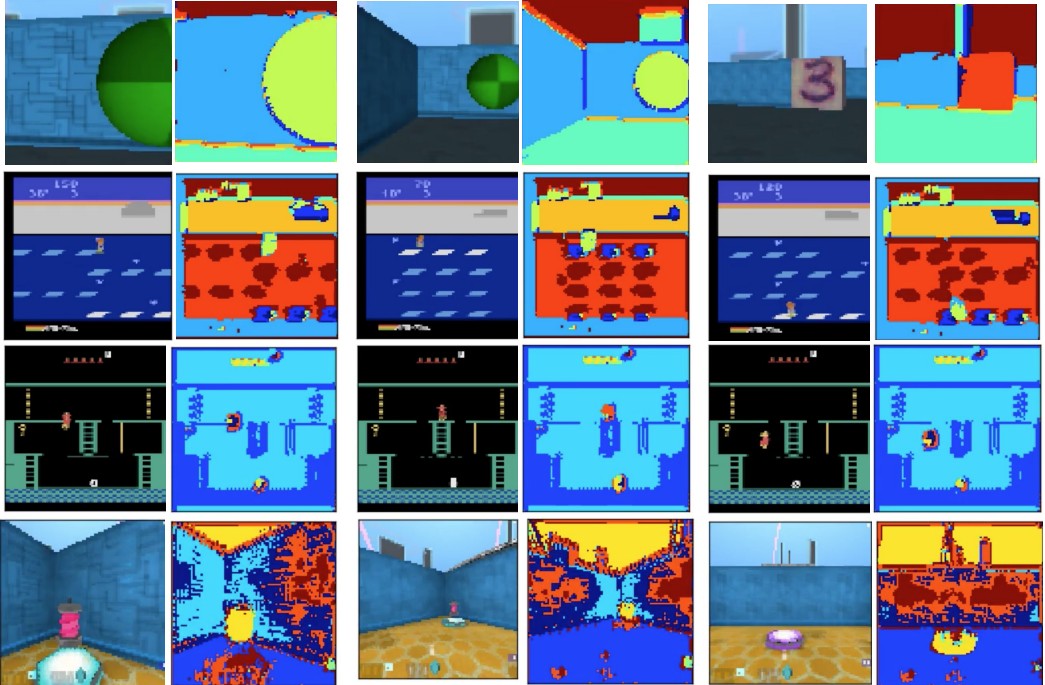

Figure 3: **Unsupervised Visual Abstractions:** Inputs and inferred visual abstractions using our model. Each row represents temporally close frames from different environments including navigation and Atari. The colors in the abstraction images correspond to visual instances with temporally consistent labels.

where $R_\tau = \sum_{t=\tau}^{\tau+\mathbf{U}-1} \gamma^{t+\tau} r_t + \gamma^{t+\mathbf{U}} \max_{a'} Q^{\text{task}}(s_{\tau+\mathbf{U}}, a'; \theta_{task})$, where $\mathbf{U}$ is the length of the unroll. The loss function $\mathcal{L}_{meta}$ and all the $\mathcal{L}_{bank}^{e,m}$ can be written in a similar fashion. In our experiments we use $Q(\lambda)$ and learn all parameters with stochastic gradient descent by sampling experience trajectories from the shared replay buffer (see Algorithm 1 for details).

## 4 EXPERIMENTS

The implementation has a learning setup inspired by the batched actor critic agent (Espeholt et al., 2018), with a central GPU learner and multiple actors (64 in most experiments). See 6 for details regarding visual abstraction architecture and agent network architecture.

**The exploration setup.** For the baseline we have 3 types of actors differing only in the exploration parameter corresponding to *high exploitation*, *high exploration* and *medium exploration*. From the total number of 64 actors that corresponds to 20, 10 and 34 respectively with epsilon values of .001, .5, .01. Both the meta agent and base policies require exploration so we keep the same split but the *high exploration* is considered independently for the base and meta policies i.e. half the actors have $\epsilon_{meta} = .5$ and $\epsilon_{base} = .01$ and half have $\epsilon_{meta} = .1$ and $\epsilon_{base} = .5$. Furthermore we split equally the medium exploration actors in 3 groups with ($\epsilon_{meta} = .01, \epsilon_{base} = .1$), ($\epsilon_{meta} = .33, \epsilon_{base} = .001$) and ($\epsilon_{meta} = .01, \epsilon_{base} = .1$) respectively. This provides much more stable learning at both the higher and lower levels of the behavior hierarchy.

### 4.1 NAVIGATION

We tested our approach on a 3D navigation domain with sparse rewards and hard exploration requirements. The domain, whose top down view (not visible to the agent) is shown in figure 4, contains four rooms each having a textured number at the entrance. The agent receives the image observation as well as *hint*, a number from 1 to 4, which indicates the number of the room which contains the target object, a green sphere. The goal is to reach the target as often as possible in the

---

**Algorithm 1** Learning algorithm

---

1: Inputs: $N$ number of episodes, $\epsilon_{base}$ and $\epsilon_{meta}$ exploration parameters, $\mathbf{T}$ the commitment length, $\lambda_{task}, \lambda_{meta}, \lambda_{bank}$ cost function parameters
2: Initialize experience replay buffer $\mathcal{D}$ and parameters $\{\theta_{meta}, \theta_{task}, \theta^{1,1}, ..., \theta^{e,m}\}$ for the meta-control agent, task agent and options models respectively.
3: **for** $i = 1$ to $N$ **do**
4:     Initialize environment and get start state $s$
5:     $s_0 \leftarrow s$
6:     **while** $s_t$ is **not** terminal **do**
7:         **if** $t \equiv 0 \mod \mathbf{T}$ **then**
8:             $a_t^{\text{meta}} = (e_t, m_t) \leftarrow \text{EPSGREEDY}(s_t, \epsilon_{meta}, \theta_{meta})$
9:         **end if**
10:        Compute abstract features $v_t$ from $s_t$ (Section 3.1).
11:        Compute intrinsic rewards $\mathbf{r}_{int} = (r^{e,m} | \forall e \leq \mathbf{E}, m \leq \mathbf{M})$ from $v_t$ and $v_{t-1}$
12:        **if** $e_t \leq \mathbf{E}$ **then**
13:            $a_t \leftarrow \text{EPSGREEDY}(s_t, \epsilon_{base}, Q^{e_t, m_t})$       % selected $Q$ function from bank
14:        **else**
15:            $a_t \leftarrow \text{EPSGREEDY}(s_t, \epsilon_{base}, Q_{task})$         % selected $Q^{\text{task}}$
16:        **end if**
17:        Execute $a_t$ and obtain next state $s_{t+1}$ and extrinsic reward $r_t$ from environment
18:        Store transition $(s_t, (e_t, m_t, a_t), r_t, s_{t+1})$ in $\mathcal{D}$
19:        Use RMSProp to optimize $\mathcal{L}_{abs}(\theta_{abs})$         %see eq. (2))
20:        Use RMSProp to optimize $\lambda_{task}\mathcal{L}_{task} + \lambda_{meta}\mathcal{L}_{meta} + \lambda_{bank}\frac{1}{\mathbf{ME}}\sum_{e=1}^{\mathbf{E}}\sum_{m=1}^{\mathbf{M}}\mathcal{L}_{bank}^{e,m}$
    (see eq. (4))
21:     **end while**
22: **end for**

---

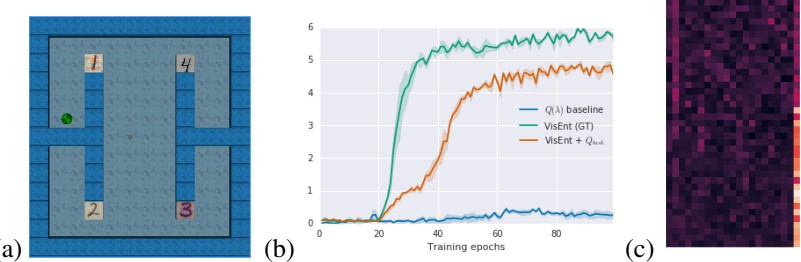

(a)               (b)                 (c)

Figure 4: **Navigation domain.** (a) Top-down view of a 3D Maze. The agent observes pixels from the first person view. The task is described in section 4.1. (b) The green curve denotes an optimal agent with access to ground truth visual abstractions. Our learnt model achieves close to optimal performance while the baseline fails to solve the task. (c) Plot showing how the meta control policy switches between options bank and task control policies. Time increases top to bottom. The right most column is the task policy and all other columns denote option policies. Initially the agent uses the options policies to explore and then gradually shift over to the task policy as training progresses.

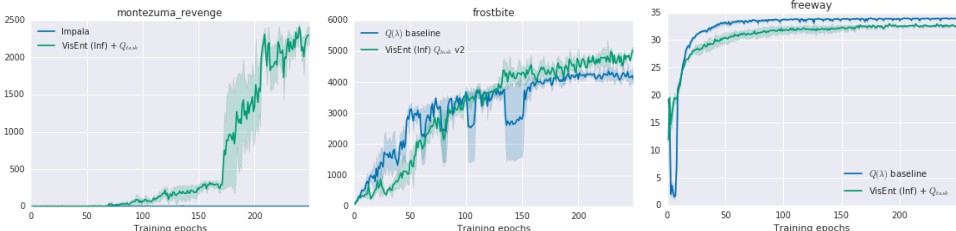

Figure 5: **Quantitative results on the Atari domain:** Average returns per episode on 3 Atari games where exploration is considered very challenging. On these domains we achieve better or comparable results to a strong baseline.

limited time budget (250 steps). Once the goal is reached the target is relocated in another randomly chosen room and the hint is updated accordingly. To solve the task the agent needs to associate the hint with its position in the environment, which in turn means exploring far away regions in the maze because the target may move in a far away place. The policy bank provides exactly this type of exploration basis set. A baseline agent with the same architecture but without the policy bank cannot solve this task in 100M steps see Figure 4. In this domain we can also query the environment representation to determine if a given pixel comes from either a wall, the skyline, floor, one of the textured numbers or the target sphere. An agent with our proposed architecture but computing intrinsic rewards based on this priviledged information solves the tasks easily. Most interestingly, an agent that uses the abstraction inference method we propose can also solve it too, though with slightly worse performance.

To gain an insight into the evolution of the agent policy, we plot in Figure 4 (c), a histogram of the $Q^{\text{meta}}$ policy actions during training (time is on the vertical axes and flows from top to bottom). The leftmost 20 actions represent the policy bank and the rightmost one represents $Q^{\text{task}}$. We can see that the agent uses most of the policies in the bank but as training progresses learns to rely more and more on $Q^{\text{task}}$ which is the optimal behavior.

## 4.2   ATARI

We have run our agent with the same architecture and parameters on hard exploration Atari games where we can show that our agent has better performance than the baseline with a comparable architecture and losses (see Figure 5).

## 4.3   DM LAB

In order to see if our model scales to more visually challenging environments we have run our agent on 3 varied "DMLab-30" levels(Beattie et al., 2016). We show the training curves as a function of time in Figure 6. In the "non-match" task the agent teleports first to a room with one object then a room with two, one of which is the same as before. To get a positive reward the agent has to move on top of the other object, otherwise it receives a negative reward. The sequence continues for a fixed number of steps. Humans achieve 66 points and a state of the art agent gets 26. Our $Q$ learner baseline achieves only 9 points whereas our proposed agent achieves 33. Though this is meant to be a memory task structured exploration seems to help achieve much better scores than the baselines.

We have also considered an experiment on the "keys doors" task. In this task the agent has to successively pickup keys to unlock doors which seems like a task structure were our representation could be more effective than the baseline. We found that, though both methods are competitive, our representation was not sufficent to learn a better policy. We think that this may be due to noise in the abstraction inference as well as a planning aspect that is not well enough handled in our agent in its current form. Finally, our agent is outperformed by the baseline on the challenging watermaze task. This is most likely due to the policy bank exploring mostly straight trajectories rather than circular ones which are more appropriate on this task.

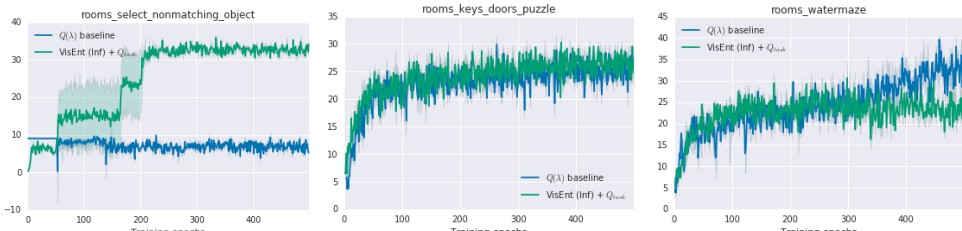

Figure 6: **Quantitative results on the DMLab domain:** Average returns per episode on 3 DMLab levels.

## 5 DISCUSSION

We have shown that it is possible to design unsupervised structured exploration schemes for model-free DRL agents, with competitive performance on a range of environments given just raw pixels.

One of the biggest open question moving forward is to find strategies to balance structure or inductive biases and performance. Our current solution was to augment the meta-controller with $Q^{\text{task}}$ along with the options bank as sub-behaviors. The typical strategy that agents follow is to rely on the options bank early in training and then use this experience to train the $Q^{\text{task}}$ policy for optimality as training progresses. This is reasonable given that the options models may not cover the optimal policy but could serve as a good exploration algorithm throughout training. As new unsupervised architectures and losses are discovered, we expect to narrow the gap between the optimal desired behaviors and the options bank.

Learning visual entities from pixels is still a challenging open problem in unsupervised learning and computer vision. We expect novel sampling schemes in our proposed architecture to improve the entity discovery results. Other unsupervised video segmentation algorithms and discrete latent variable models could also be used to boost the discovery process.

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

# 6 SUPPLEMENTARY MATERIAL

**Visual Abstraction Architecture.** The encoder for the abstractions is a set of 3 convolutional layers $3 \times 3$ kernels with 64 features each followed by ReLU nonlinearities. After a $1 \times 1$ convolutional layer with 8 outputs the features are $l_2$ normalized. We found that to work better in practice with the VQ layer sitting on top. The VQ layer has 8 elements unless otherwise stated. Since the strides are always 1 the VQ output has the same spatial dimensions as the input image. The global loss is a stack of convolutions $3 \times 3$, followed by a $2 \times 2$ max pooling with stride two to reduce the resolution of the output and then a ReLU non-linearity. The output is then flattened to give the global embedding of the abstract representation. The image embedding is a two-layer 8 filter convnet with a ReLU non-linearity inbetween.

**Agent architecture.** The agent architecture is a standard 3 layer convolutional stack similar to Mnih et al. (2015) with 512 hidden unit output and an LSTM on top. The output of the LSTM is fed into an *visual abstraction* selection Q function, a *measurement* selection Q function, a regular *task* Q function layer and the *policy bank* i.e. a layer with $\mathbf{M} \times \mathbf{E} \times \text{num\_actions}$ outputs. The Q function layers are all dueling heads as in Wang et al. (2016).

**Setup and baseline.** Our setup is very similar to the one in (Espeholt et al., 2018). Multiple actors (64 in most examples) in the acting loop that send trajectories to a shared learner which processes them in batched fashion (batch size of 32). The main difference is the use of value based $Q(\lambda)$ loss (with a $\lambda$ value of .85) instead of actor-critic with off-policy correction. The baseline agent has the same exact architecture and loss as our agent. In fact if we ignored the meta control $Q$ function and the options bank we get our baseline exactly.

