# OpenReview forum: "Learning to Control Visual Abstractions for Structured Exploration in Deep Reinforcement Learning"
_ICLR.cc/2019/Conference_

### Official Review · AnonReviewer3 · 2018-11-01
**Insufficient clarity**

**Rating:** 4
**Confidence:** 3

**Review:**

REVISION: thanks for the clarification. I have slightly increased my rating (to 4).

This paper tackles a very interesting subject but lacks sufficient clarity of presentation to allow me to do a proper review.

First, there are many sentences which are not well-formed or are ambiguous (in pretty much all the sections). Then there are terms which are introduced without being first clearly explained or defined. Finally, there are issues with the mathematical clarity as well, with many notations which are used without being explained or defined. Sometimes one can figure out the missing information later (e.g., fig 1 talks about mutual information objectives without stating if we want to maximize or minimize it, but later in the text we figure that out) but it makes reading very difficult.

What is a 'transformed one' (on page 2)
What is a 'geometric intrinsic reward'?
Where are the intrinsic rewards defined?
What is a 'non-parametric classifier'? A neural net? an kernel SVM?

There are also some mathematical problems:
- if f (page 3) has a discrete output, then it will probably lose information, so it cannot be inverted (contrary to the stated assumption that f(x)!=f(y) for x!=y)
- what are the differences between the different Q functions being defined? do the correspond to different action spaces? What is Q_task? What is pi_meta?
- in eqn 2, I do not think that the log q_c term maximizes the mutual information between actions and (G(t),G(t+1)), i.e. it would be missing an entropy term
- what is Z_c in eqn 2?

---

> ### Author Response · Authors · 2018-11-22
> **Significantly improved the clarity of our writing in the revision**
>
> Thanks a lot for the critical feedback. We have improved the clarity of our writing and made the contributions clearer. We urge you to read the revised paper and hopefully it will convince you of our contributions.
>
> Q: “What is a 'transformed one' (on page 2)”
> The original image is transformed by applying the following operators: additive color changes in HSV space, horizontal flips and spatial shifts.
>
> Q: “What is a 'geometric intrinsic reward?”
> We compute geometric measurements as position_x, position_y, area of each segment. Temporal changes in these measurements gives our agent a notion of how it moves relative to extracted entities. If an entity’s area becomes larger/smaller it means it is getting closer/farther away from it. Because both getting closer and farther may be useful we train to maximize both the measurements and its negative. Because these are over segments trained in an unsupervised way it makes them intrinsic.
>
> Q: “What is a 'non-parametric classifier'? A neural net? an kernel SVM?”
> We use a K-NN classifier with a distance metric induced by a neural network. We apologize for not explaining this clearly and have updated Sec 3.1 to reflect it more clearly.
>
> Q: “if f (page 3) has a discrete output, then it will probably lose information, so it cannot be inverted (contrary to the stated assumption that f(x)!=f(y) for x!=y)”
> Sorry for the confusing explanation. We have updated Sec 3.1. The VQ layer will only keep information that is important for distinguishing states as per space-time consistency and controllability constraints, while discarding other information.
>
> Q: “what are the differences between the different Q functions being defined? do the correspond to different action spaces? What is Q_task? What is pi_meta?”
> Q_meta and Q_task optimize the task reward while Q^{e,m} optimized the m-th reward of the e-th entity. Q_task and all Q^{e,m} are defined over the environment actions but Q_meta’s action space is ExM+1 as described at the end of section 3. Pi meta is the policy derived from the corresponding Q, in our case this is epsilon-greedy.
>
> Q: “what is Z_c in eqn 2?”
> This is a new random variable that we have to introduce in order to have a well defined objective. This has been explained more clearly in the updated text.

---

### Official Review · AnonReviewer1 · 2018-11-02
**A  Structured Exploration Algorithm with Visual Abstractions**

**Rating:** 5
**Confidence:** 3

**Review:**

This paper proposed an algorithm for structured exploration in deep reinforcement learning via learning the visual abstractions from pixels. The proposed method learns discrete visual abstractions and derives intrinsic reward functions from them so as to help the agent to optimize the policy.

The proposed method is interesting in that learning the visual abstractions together with the policy may assist in computing an optimal policy. The method is learning a meta Q function and (E * M+1) other Q functions. The authors mentioned that their work is most similar to hierarchical-DQN (Kulkarni et al., 2016) but this work required hand-crafted instance segmentation and the agent architecture do not learn about many intrinsic rewards learners. However, I am concerned if the proposed method solved the problem with the need of hand-crafted instance segmentation since, as shown in the Algorithm 1 and the caption of Figure 2, Q_{meta} acts every T steps. I do not understand why the meta Q function is used to propose actions for every fixed number of steps. Besides that, though the proposed method does have many intrinsic reward functions (in fact, there are E * M additional intrinsic reward functions). However, the authors did not show in the experiments if having too many intrinsic reward functions helps a lot. It will be better if the authors can show that, larger values for E or M can make the performances better.

Another concern I have is on some of the experiment results. For the experiment results in Figure 5 and 6, only in the left figures can the results of the proposed methods outperform the baselines. Besides that, the authors may need to describe the baseline methods in the experiments in more details.

Also, it will be better if the authors can improve the paper a little bit with the writing. For example, it will be better if the authors can explain the variables X, Y and the distribution q when mentioning Equation 1 so that it is easier to understand the paper. Also, there are some typos, such as the section reference on line 10 of Algorithm 1, the definition of the g function on the last line of page 3 (I guess the authors want to write "{0...E}" instead of "{0, E}") and the second sentence of the experiment section (at least I did not see the supplementary sections, but the authors mentioned that). It is better if these typos can be fixed.


References:

Tejas D Kulkarni, Karthik Narasimhan, Ardavan Saeedi, and Josh Tenenbaum. Hierarchical deep reinforcement learning: Integrating temporal abstraction and intrinsic motivation. In Advances in neural information processing systems, pp. 3675–3683, 2016.

---

> ### Author Response · Authors · 2018-11-22
> **Addressed concerns about the experimental setup/experiments and clarity of the ideas/contributions**
>
> Thanks a lot for all the feedback and suggestions. This has helped us improve the clarity of our writing.
>
> Q: “However, I am concerned if the proposed method solved the problem with the need of hand-crafted instance segmentation since, as shown in the Algorithm 1 and the caption of Figure 2, Q_{meta} acts every T steps”
>
> We do not require hand-crafted instance segmentation, such as in Kulkarni et al. We use an information theoretic loss to learn visual abstractions, which are further used to learn temporal abstractions or options. Q_meta picks an internal action, i.e. indexes into either the options bank or Q_task, and is optimized to maximize the environment task reward. This choice is fixed for T steps (option termination condition) and the chosen sub-controller executes real actions in the environment.
>
> Q: “However, the authors did not show in the experiments if having too many intrinsic reward functions helps a lot. It will be better if the authors can show that, larger values for E or M can make the performances better.”
> We experimented with large E and found that the model discovers similar number of entities while often predicting empty segments. In terms of M, we have reported the simplest set of measurements that capture the 2D and 3D temporal structure of different environments. There could be many more functions which should be explored building upon this work.
>
> Q: “Another concern I have is on some of the experiment results … “
> We are using the Espeholt et al. training setup but with Q(lambda) to make it comparable with our agent. We have clarified this in the revised supplemental. Our method either outperforms or is in the same ballpark as the baseline. In tasks with sparse rewards, our method is especially beneficial as the options bank aids temporally extended exploration. Our main claim is that prior DRL agents have not been able to  object based structured exploration from pixels. Scaling this approach and making it more robust is an open question but we believe we have shown a promising avenue along these lines.

---

> > ### Comment · AnonReviewer1 · 2018-11-23
> > **Option termination condition**
> >
> > Thanks for the rebuttal. Can you explain why the option termination condition is that Q_{meta} picks an action for every fixed number of steps? Why the termination condition is independent with the task, the environment and the state? Why this is a good choice?

---

> > > ### Author Response · Authors · 2018-11-24
> > > **termination condition**
> > >
> > > Thanks for this question. The options in our setting maximize or minimize entity attributes, so there isn't a natural goal success criteria (e.g. sometimes there could be obstacles in an entity's path or none at all). In some cases it might be possible to make statements about goal achievement, for instance if the agent can learn to reason about immovable obstacles. This could be an interesting direction to explore in the future. Also in other papers which have considered adaptive T in the setting of relational goals (e.g. Kulkarni et al.), an internal goal critic could clearly measure goal success contrary to our goal space. In practice, a single T (=20) works well across all our experiments and domains.

---

### Official Review · AnonReviewer2 · 2018-11-04
**The paper is unfortunately written quite confusingly such that it is hard to evaluate the contribution of the potentially interesting ideas.**

**Rating:** 4
**Confidence:** 3

**Review:**

The appproach introduces visual abstractions that are used for reinforcement learning. The abstractions are learned using a lower bound on the mutual information and options are created to generate different measurements for each abstraction. The algorithm hence learns to "control" each abstraction as well as to select the options to achieve the overall task. The algorithm is tested on a 3D navigation task and a few Atari tasks which are known for difficult exploration.

The paper might contain some interesting ideas, however, I am quite confused about the paper due to lack of clarity in writing. The approach is not properly motivated, many equations are not really eplained and important information is missing, so it is really hard to evaluate the contribution of the approach. Please see below for more comments:
- It is unclear how the intrinsic reward is defined (which is critical to understand the approach).
- It is unclear what the M different measurements are or for what they are used for.
- It is unclear qhy equation 1 defines a classification loss. Distribution q is not defined in Eq (1).
- I do not understand the description of Q-meta in caption of Figure 2, "Qmeta acts every T steps, which is the fixed temporal
commitment window, and outputs an action to select and execute either: (1) composition over Q
function from the option bank indexed by a particular entity and an intrinsic reward function or (2)
the Qtask policy which outputs raw actions." How can an action be a composition over Q-function and a intrinisic reward function? Please clarify what Qmeta and Qtask do in the text right in the beginning.

I have to say that the paper confused me too much that it is likely I missed the point of the paper. On the positive side, I think the learning of the abstractions using lower bounds of the mutual information is very interesting. The authors should work on their presentation and this could be a very nice paper.

---

> ### Author Response · Authors · 2018-11-22
> **Improved clarity of the writing, clearly wrote down our contributions and made revisions to the paper**
>
> We want to thank the reviewer for valuable feedback in improving the clarity of the paper.
> Q: “It is unclear how the intrinsic reward is defined (which is critical to understand the approach).”
> We apologize for the confusion. We mentioned this in fig 2 and sec 3 but need to make it clear. The intrinsic rewards are geometric properties of the learnt segmentations (min/max of area, centroid x, centroid y for each learnt segment). The segments are obtained directly from the spatial VQ layer. We have updated the write up to make this clear in TODO.
>
> Q: “It is unclear what the M different measurements are or for what they are used for.”
> Controlling the geometric features of learnt segments is a principled way to learn skills to control different object attributes (relative distance to observer, relative position of objects) in 2D/3D scenes. The M different measurements are the affine variables (e.g. position_x, position_y, area) of each segment. Controlling such geometric features can enable higher levels of behaviors such as reaching towards an object (max area), avoiding certain objects (min area), moving an object away towards the left (min position x), controlling the avatar’s position on the screen etc. We have reflected this in Sec 3 and the introduction.
>
> Q: “It is unclear why equation 1 defines a classification loss. Distribution q is not defined in Eq (1)”.
> We apologize for not clearly explaining this. This is a classification loss due to reasons and derivations explained in prior work, namely -- MINE and CPC. We now state this in Sec 3.2. We have also defined q in eq (3) in the revised draft.
>
> Q: “ Please clarify what Qmeta and Qtask do in the text right in the beginning.“
> Q_meta picks an internal action, i.e. indexes into either the options bank or Q_task (gets extrinsic reward and operates over low level actions), and is optimized to maximize the environment task reward. This choice is fixed for T steps and the chosen sub-controller executes real actions in the environment. We added a clear explanation in the introduction as well as Sec 3.

---

### Author Response · Authors · 2018-11-22
**Rebuttal summary**

We want to thank all reviewers for their critical feedback and suggestions, which has already helped us improve the paper’s clarity and presentation. All the reviewers agree that the paper tackles an interesting and important problem of discovering spatial and temporal abstractions given raw observations and actions. The main concerns were about the clarity of the writing, making it hard to clearly assess the underlying contributions.

We have significantly improved the presentation of our ideas considering all the feedback and explicitly made our contributions clearer. Our two key contributions are: (1) An information theoretic loss and architecture to learn spatio-temporal visual abstractions given raw pixels and actions, (2) a new agent architecture which learns temporal abstractions grounded in the geometry of the discovered visual abstractions. There have been several agent architectures in the past that make use of object-oriented information for constructing states and to aid exploration. However, this is the first agent architecture that simultaneously learns visual and temporal abstractions, while demonstrating clear improvements over baselines on hard 3D navigation and Atari games.

We urge all reviewers to read the updated version of the paper, as we have carefully addressed and incorporated all critical feedback and suggestions.

---

### Meta-Review · Area_Chair1 · 2018-12-15
**Interesting structured exploration idea, not clear nor detailed enough**

**Confidence:** 4
**Recommendation:** Reject

**Metareview:**

The paper presents an unsupervised visual abstraction model, used for reinforcement learning tasks. It is trained through intrinsic rewards, generated from temporal differences of inputs. This is similar to "learning to control pixels". The method is tested in DM Lab (3D environment, 2D navigation tasks) and Atari (Montezuma's Revenge).

The paper is at times hard to follow, and it seems the improvements accompanying the rebuttals did not convince reviewers to change their notes significantly. The experiments do not contain enough comparisons to other models, baselines, nor ablations, to sustain the claims.

In its current form, this is not acceptable for publication at ICLR.